**Data Availability Statement:** All data files are available from the COVID-19 database Sharing/BR repository (FAPESP. COVID-19 DataSharing/ BR,

# Long-term respiratory follow-up of ICU hospitalized COVID-19 patients: Prospective cohort study

**Carlos Roberto Ribeiro Carvalho**[ID][1]*, **Celina Almeida Lamas**[1], **Rodrigo Caruso Chate**[2], **João Marcos Salge**[1], **Marcio Valente Yamada Sawamura**[2], **André L. P. de Albuquerque**[1], **Carlos Toufen Junior**[1], **Daniel Mario Lima**[ID][3], **Michelle Louvaes Garcia**[1], **Paula Gobi Scudeller**[1], **Cesar Higa Nomura**[2], **Marco Antonio Gutierrez**[3], **Bruno Guedes Baldi**[ID][1], **HCFMUSP Covid-19 Study Group**[¶]

1 Pulmonary Division, Heart Institute (InCor), Hospital das Clínicas, Faculdade de Medicina, Universidade de São Paulo (HCFMUSP), Sao Paulo, SP, Brazil, 2 Radiology Institute (InRad), Hospital das Clínicas, Faculdade de Medicina, Universidade de São Paulo (HCFMUSP), Sao Paulo, SP, Brazil, 3 Informatics Division, Heart Institute (InCor), Hospital das Clínicas, Faculdade de Medicina, Universidade de São Paulo (HCFMUSP), Sao Paulo, SP, Brazil

¶ Membership of the HCFMUSP Covid-19 Study Group is provided in the Acknowledgments.
* carlos.carvalho@hc.fm.usp.br

## Abstract

### Background

Coronavirus disease (COVID-19) survivors exhibit multisystemic alterations after hospitalization. Little is known about long-term imaging and pulmonary function of hospitalized patients intensive care unit (ICU) who survive COVID-19. We aimed to investigate long-term consequences of COVID-19 on the respiratory system of patients discharged from hospital ICU and identify risk factors associated with chest computed tomography (CT) lesion severity.

### Methods

A prospective cohort study of COVID-19 patients admitted to a tertiary hospital ICU in Brazil (March-August/2020), and followed-up six-twelve months after hospital admission. Initial assessment included: modified Medical Research Council dyspnea scale, SpO$_2$ evaluation, forced vital capacity, and chest X-Ray. Patients with alterations in at least one of these examinations were eligible for CT and pulmonary function tests (PFTs) approximately 16 months after hospital admission. Primary outcome: CT lesion severity (fibrotic-like or non-fibrotic-like). Baseline clinical variables were used to build a machine learning model (ML) to predict the severity of CT lesion.

### Results

In total, 326 patients (72%) were eligible for CT and PFTs. COVID-19 CT lesions were identified in 81.8% of patients, and half of them showed mild restrictive lung impairment and impaired lung diffusion capacity. Patients with COVID-19 CT findings were stratified into two

2021. Available: https://repositoriodatasharingfapesp.uspdigital.usp.br).

**Funding:** The Funding Statement that should be included on the Funding Statement section of the online submission form is: "MAG acknowledge the Sao Paulo Research Foundation for financial support (grants number 16/17078-0 and 14/50889-7). The funders had no role in study design, data collection and analysis, decision to publish, or preparation of the manuscript.

**Competing interests:** The authors have declared that no competing interests exist.

categories of lesion severity: non-fibrotic-like (50.8%-ground-glass opacities/reticulations) and fibrotic-like (49.2%-traction bronchiectasis/architectural distortion). No association between CT feature severity and altered lung diffusion or functional restrictive/obstructive patterns was found. The ML detected that male sex, ICU and invasive mechanic ventilation (IMV) period, tracheostomy and vasoactive drug need during hospitalization were predictors of CT lesion severity(sensitivity,0.78±0.02;specificity,0.79±0.01;F1-score,0.78±0.02;positive predictive rate,0.78±0.02; accuracy,0.78±0.02; and area under the curve,0.83±0.01).

## Conclusion

ICU hospitalization due to COVID-19 led to respiratory system alterations six-twelve months after hospital admission. Male sex and critical disease acute phase, characterized by a longer ICU and IMV period, and need for tracheostomy and vasoactive drugs, were risk factors for severe CT lesions six-twelve months after hospital admission.

## 1. Introduction

Severe acute respiratory syndrome coronavirus 2 (SARS-CoV-2) has spread worldwide since the end of 2019. SARS-CoV-2 infection triggered the Coronavirus Disease 2019 (COVID-19) pandemic, which has caused more than six million deaths globally to date [1]. Scientists worldwide have made efforts to clarify the clinical consequences and prognosis of the acute phase of COVID-19 [2–4]. However, studies assessing the long-term consequences of this disease, especially considering the recovery of critically ill patients in the intensive care unit (ICU), are still scarce. The World Health Organization (WHO) has drawn attention to long COVID, which is the persistence of disease-related symptoms for more than three months after recovery [5–7]. Frequent manifestations of long COVID include dyspnea, fatigue, fever, myalgia, headache, and fibrotic-like lung abnormalities [5, 7–9].

A recent cohort study by the United States Department of Veterans Affairs showed that hospitalized ICU patients had a higher risk of death and pulmonary disease than non-ICU patients six months after COVID-19 infection [10]. Our previous study showed that 76.5% of patients who had recovered from COVID-19 still had at least one abnormality on chest computed tomography (CT) six months after hospital admission, which was more frequent in ICU than in ward patients [11]. Thus far, most cohort studies on recovered ICU COVID-19 patients have focused on long-term symptoms than pulmonary assessments. In view of this, a recent Dutch cohort study on 246 patients one year after COVID-19 ICU treatment demonstrated that 74% of patients still reported physical symptoms, 26% reported mental symptoms, and 16% reported cognitive symptoms. Moreover, a larger study that evaluated 390 patients six months after recovery from COVID-19 in China was restricted to only 4% of ICU patients [12]. Most of those ICU patients required invasive mechanical ventilation (IMV), had more comorbidities, and worse lung function, therefore, further studies are needed for a better insight.

It is well known that ICU hospitalization has inherent risk factors that could lead to future problems even in the non-COVID-19 context [13]. A study of survivors of other diseases caused by viruses of the SARS family showed that pulmonary sequelae can persist for up to 15 years [14], in addition to being correlated with a longer duration and severity of the acute phase of the disease, and consequently, the need for ICU hospitalization [15]. The post-

recovery effects included reduced diffusion capacity for carbon monoxide (DLCO), restrictive pattern in pulmonary function tests (PFTs), and ground glass opacities on computed tomography (CT) scan [16, 17].

These previous results reinforce the well-known assumption that ICU hospitalization represents a risk factor for development of lung abnormalities in the long term and provide clues that ICU hospitalization due to COVID-19 could lead to chronic lung CT damage, which should be investigated. Thus, it is essential to provide insights regarding ICU hospitalization that could influence the persistence of respiratory system alterations, considering that the COVID-19 pandemic is ongoing and the possibility of ICU hospitalization still exists. Therefore, we aimed to evaluate the respiratory outcomes of a consecutive large cohort of patients admitted to ICUs for COVID-19, focusing on the assessment of lung imaging and PFTs, and to determine the risk factors in the acute phase of the disease that could be predictors of chronic lung injury.

## 2. Materials and methods

### 2.1. Study design and participants

This was a prospective cohort study of COVID-19 patients admitted to the ICUs of Hospital das Clínicas, Faculdade de Medicina, Universidade de São Paulo (HCFMUSP), São Paulo, Brazil, from March 30 to August 31st, 2020. HCFMUSP was a reference center for the treatment of critically ill COVID-19 patients in Brazil, with 300 ICU beds, and had adopted an institutional treatment protocol. The protocol included a protective ventilation strategy (tidal volume < 8 ml/kg and plateau pressure < 30 cm $H_2O$), specific pharmacological treatment, thrombosis prophylaxis, and sedation [18]. Six to twelve months after hospital admission, patients aged 18 years who had RT-PCR-confirmed SARS-CoV-2 infection during hospitalization were consecutively invited to participate in the study.

This study was part of a large protocol previously described [19] and was approved by the Research Ethics Committee of our institution (No. 31942020.0.000.0068). Written informed consent was obtained from all the patients. HCFMUSP electronic medical records were assessed for the retrospective collection of patients' hospitalization clinical data, during the acute phase of the disease, such as comorbidities, symptoms, smoking history, length of ICU stay, and IMV parameters during the first 24 h of mechanical ventilation. All clinical data were cataloged in a structured form using REDCap software (https://www.redcapbrasil.com.br/).

### 2.2. Follow-up protocol

The follow-up visit procedures have been previously described [19]. Patients underwent a face-to-face general evaluation during the follow-up that included anthropometric examination, and an initial pulmonary assessment, including the modified Medical Research Council (mMRC) dyspnea scale, oxygen saturation ($SpO_2$) measured by pulse oximetry at rest and after the 1-min sit and stand test, spirometry, and chest X-ray (CXR) [19]. The protocols used to perform these tests have been described previously [11, 19]. The results of CXR images were evaluated as: normal/with findings not related to COVID-19 (cardiomegaly and pulmonary nodules, for instance) or findings probably related to COVID-19 (bilateral linear and/or reticular opacities, especially peripheral opacities) [11]. Two thoracic radiologists who were blinded to the particulars of the study evaluated the chest CXR images independently. Disagreements were resolved through consensus.

Based on the general evaluation results, patients who met at least one of the following criteria were enrolled to undergo chest CT and PFTs during a second complementary face-to-face evaluation: (a) mMRC≥2; (b) resting $SpO_2 \leq 90\%$ and/or a decrease in $SpO_2$ of $\geq 4\%$ during

the 1-min sit and stand test; (c) CXR findings probably related to COVID-19; and (d) forced vital capacity (FVC) < lower limit of normal (LLN) [11, 19].

The protocol used to perform the chest CT was previously described [11, 19]. Two thoracic radiologists who were blinded to the particulars of the study evaluated the chest CT images independently. Disagreements were resolved through consensus. Patients with COVID-19 CT findings were stratified into two categories according to lesion severity: with fibrotic-like changes (presence of traction bronchiectasis and architectural distortion) and without fibrotic-like changes (ground-glass opacities and reticulations) (modified from Han et al. [20]. The extent of lung involvement in these groups was quantified according to the following scores for each pulmonary lobe: 0, none; 1, <5%; 2, 5–25%; 3, 26–50%; 4, 51–75%; and 5, >75%. The total score was the sum of the scores of the five lobes, ranging from 0 to 25 [11].

PFTs were performed according to the recommendations of the American Thoracic Society [21]. The following parameters were determined: total lung capacity (TLC), forced vital capacity (FVC), forced expiratory volume in 1 s ($FEV_1$), $FEV_1$/FVC ratio and DLCO. A restrictive pattern, an obstructive pattern, and impaired diffusion capacity were defined as TLC < LLN, $FEV_1$/FVC ratio < LLN, and DLCO < LLN, respectively [22–24].

### 2.3. Data analysis

The D'agostino-Pearson test was used to determine the variables normality. Normally and nonnormally distributed data were expressed as the mean and standard deviations or median and interquartile range, respectively. The Student's t-test and MannWhitney U test were used to compare normally and non-normally distributed continuous variables, respectively. Numbers with percentages were used to describe the categorical variables, and were compared using the X2 test. The following software was used to perform the analysis: Excel 2016; Python 3.8.11; extension packages: Pandas 1.0.1; Numpy 1.19.5; Scipy 1.5.4; Scikit-Learn 0.24.0.

A Machine Learning model (ML) was developed to predict lesion severity after six to twelve months from ICU admission for COVID-19, based on baseline clinical variables. The ML prediction model was based on XGBoost, which makes use of a type of gradient boosting, where multiple decision tree models are trained in succession, each tending to improve performance. The variables collected at baseline with p<0.05 between two categories of CT lesion severity (without fibrotic-like changes and with fibrotic-like changes) were used as input variable into the ML model: sex (%), ICU length of stay (days), tracheostomy (%), duration of IMV (days) and the use of vasoactive drug (%). The ML analysis outcome was the prediction of lesion severity on CT images six to twelve months from ICU admission for COVID-19, based on baseline clinical variables. A three-fold cross-validation strategy was adopted for the training and validation sets. The ML prediction model performance was evaluated by the following metrics: sensitivity, specificity, F1-score, positive predictive rate, accuracy and area under the curve (AUC). The ML model is detailed in the S1 Appendix.

### 3. Results

Among the 2,290 patients hospitalized in the ICUs, 1,032 met the inclusion criteria and were eligible for this study. Fig 1 shows the flow of study participant selection.

Of the 1,032 eligible patients, a total of 453 (43.9%) underwent face-to-face general evaluation (52.54% men; median age 56.8, IQR 44.8–65.4) and were included in the study. The median time between hospital admission and general evaluation was 219 days (IQR 206–291). Hypertension (58%) was the most frequent comorbidity, and 39.1% of the patients had history of smoking. The median duration of ICU hospitalization was 10 days (IQR 6–18). In addition,

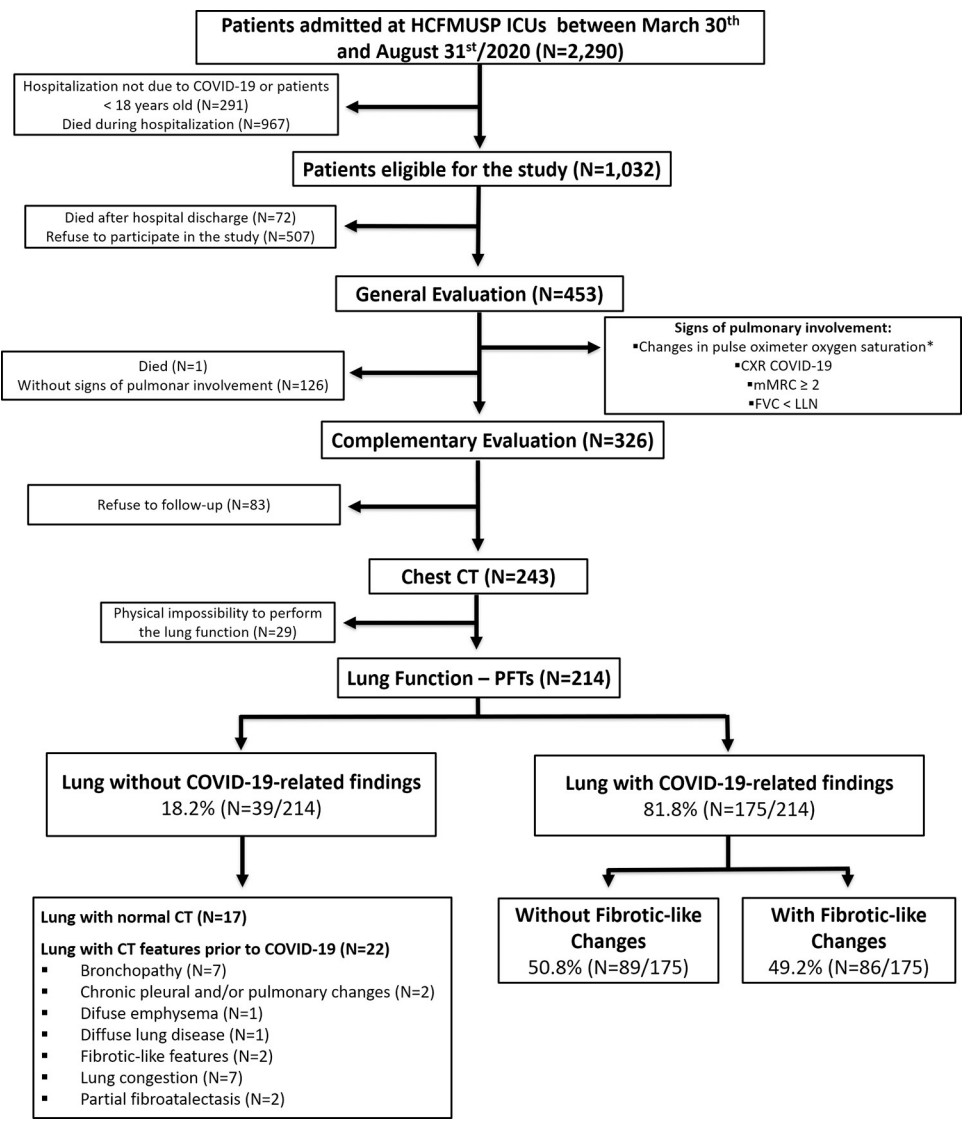

**Fig 1. Study flow chart, showing how the COVID-19 survivors were selected to participate in this follow-up until the final numbers analyzed.**

64% of these patients required IMV, with a median Simplified Acute Physiology Score 3 (SAPS3) of 57 (IQR 47–68). (Table 1 and S1 Table)

The general evaluation results showed that 72% of the patients (326 of 453) had at least one sign of pulmonary involvement. CXRs with COVID-19-related features were observed in 40.6% of patients (174 of 428 patients); FVC under the LLN was observed in 38.9% of patients (167 of 429 patients); mMRC score greater than two was observed in 29.1% of patients (131 of 450 patients), and altered oximetry was observed in 10.4% of patients (44 of 424 patients). Patients with pulmonary involvement were older, had a longer duration of ICU hospitalization, and required IMV for a longer period than those without pulmonary involvement. In addition, patients with pulmonary involvement had a slightly higher minute volume and lower compliance in the first 24h of IMV than those without pulmonary involvement. (Table 1 and S1 Table)

**Table 1. Baseline demographic and clinical characteristics of enrolled patients that underwent the general evaluation.**

| | All Patients (N = 453) | Patients with pulmonary involvement (N = 326) | Patients without pulmonary involvement (N = 127) | p-value |
|---|---|---|---|---|
| **Demographics** | | | | |
| Age, median (IQR, n)—yr | 56.8 (44.8–65.4, n = 453) | 58.4 (45.5–66.2, n = 326) | 51.1 (42.3–63.4, n = 127) | 0.004 |
| Male, % (n/N) | 52.5 (238/453) | 49.7 (162/326) | 59.8 (76/127) | 0.059 |
| BMI, median (IQR, n)—kg/m$^2$ | 28 (24.4–34, n = 418) | 28 (24–33.6, n = 304) | 28 (24.8–34.6, n = 114) | 0.268 |
| **Characteristics in ICU** | | | | |
| ICU lenght of stay, median (IQR, n)—d | 10 (6–18, n = 453) | 11 (6–19.7, n = 326) | 8 (4–14, n = 127) | <0.001 |
| SAPS 3 at admission, median (IQR, n) | 57 (47–68, n = 427) | 56 (47–68, n = 312) | 58 (46.5–68, n = 115) | 0.466 |
| D Dimer 72h, median (IQR, n)—ng/ml | 1555 (846.7–4328.2, n = 428) | 1595 (890–4608, n = 307) | 1403 (797–3799, n = 121) | 0.083 |
| CRP 72h, median (IQR, n)—mg/l | 146.8 (73.6–252.3, n = 437) | 144.8 (73.4–252.2, n = 313) | 165 (87.2–252.4, n = 124) | 0.137 |
| Dialysis, % (n/N) | 17.9 (81/453) | 17.8 (58/326) | 18.1 (23/127) | 0.822 |
| Tracheostomy, % (n/N) | 7.5 (34/453) | 9.51 (31/326) | 2.4 (3/127) | 0.009 |
| VAD, % (n/N) | 35.3 (160/453) | 35.6 (116/326) | 34.6 (44/127) | 0.913 |
| IMV during hospitalization, % (n/N) | 64 (290/453) | 65 (212/326) | 61.4 (78/127) | 0.514 |
| Duration of IMV, median (IQR, n)—d | 8 (5–13, n = 264) | 9 (6–14, n = 190) | 7 (5–11, n = 74) | 0.004 |
| **IMV at first 24 hours** | | | | |
| Tidal Volume, median (IQR, n) -ml/kg | 6.1 (5.9–6.9, n = 250) | 6.1 (5.9–6.8, n = 180) | 6.1 (5.9–7, n = 70) | 0.377 |
| Minute Volume, median (IQR, n)—l/min | 12 (10–15, n = 199) | 12 (10–15, n = 147) | 11 (10–14, n = 52) | 0.006 |
| Compliance, median (IQR, n)—mlcmH$_2$O$-1$ | 31.8 (24.5–41, n = 204) | 30 (23.7–38.7, n = 150) | 37.4 (28.2–44.3, n = 54) | 0.013 |
| Respiratory rate, median (IQR, n)—rpm | 30 (26–35, n = 220) | 30 (26–35, n = 161) | 30 (27.5–35, n = 59) | 0.273 |
| PEEP, median (IQR, n)—cmH$_2$O | 10 (8–12, n = 220) | 10 (8–12, n = 161) | 10 (8–10, n = 59) | 0.134 |
| Plateau pressure, median (IQR, n)—cmH$_2$O | 22 (19–25.5, n = 199) | 23 (20–26, n = 147) | 21.5 (19–24, n = 52) | 0.119 |
| Driving pressure, median (IQR, n)—cmH$_2$O | 12 (10–15, n = 227) | 12 (10–14, n = 59) | 12 (10–15, n = 168) | 0.137 |
| PaO$_2$/FIO$_2$, median (IQR, n)—% | 160 (124–211.5, n = 220) | 155 (120–210, n = 161) | 172 (137.5–212.5, n = 59) | 0.3 |
| Compliance ≥ 40, % (n/N) | 27.6 (63/228) | 22.6 (38/168) | 41.7 (25/60) | 0.007 |
| Plateau pressure ≥ 28, % (n/N) | 12.4 (28/226) | 13.2 (22/167) | 10.2 (6/59) | 0.65 |
| Driving pressure ≥ 15, % (n/N) | 26.9 (61/227) | 28.6 (48/168) | 22 (13/59) | 0.395 |
| PaO$_2$/FIO$_2$ ≤ 150, % (n/N) | 47.4 (121/255) | 50 (93/186) | 40.6 (28/69) | 0.205 |

Values are presented as median [IQR, n] or % (n/N). *Abbreviations*: BMI, body mass index; CRP, c-reactive protein; d, days; FIO$_2$, inspired fraction of oxygen; ICU, intensive care unit. IMV, invasive mechanical ventilation; PaO$_2$, arterial partial pressure of oxygen; PEEP, positive end-expiratory pressure; SAPS3, Simplified Acute Physiology Score 3; VAD, vasoactive drugs; yr, years.

The 326 patients who had at least one sign of pulmonary involvement on general evaluation were enrolled in a complementary evaluation with CT and PFTs (Fig 1). The median interval between general and complementary evaluations was 43 days (IQR 28–57). Chest CT and PFTs were completed in 74.5% and 65.6% of the selected patients, respectively. The demographic and clinical characteristics of the patients stratified by the completion or absence of chest CT or PFTs are described in S2 and S3 Tables, respectively.

The CT and PFT results are presented in Table 2. At least one abnormal CT feature was found in 85% of the patients. The most common abnormalities were ground-glass opacities, parenchymal bands, reticulations, traction bronchiectasis and architectural distortions.

**Table 2. Chest CT and PFTs among COVID-19 survivors at the follow-up.**

| Examinations | Results at the follow-up |
|---|---|
| **Chest CT** | **Total (N = 243)** |
| At least one abnormal CT feature, % (n) | 85 (206) |
| Ground-glass opacities, % (n) | 81 (196) |
| Parenchymal bands, % (n) | 71 (173) |
| Reticulations, % (n) | 61 (149) |
| Traction bronchiectasis, % (n) | 38 (93) |
| Architectural distortion, % (n) | 31 (75) |
| Bronchial wall thickening, % (n) | 26 (63) |
| Perilobular opacities, % (n) | 20 (49) |
| Mosaic attenuation pattern, % (n) | 20 (48) |
| Consolidations, % (n) | 1.2 (3) |
| Pneumatocele, % (n) | 1.2 (2) |
| Honeycombing, % (n) | 0 |
| **PFTs** | **Total (N = 214)** |
| FVC, mean ± SD—% of predicted | 80.8 ± 14.5 |
| FVC < LLN, % (n) | 45 (97) |
| $FEV_1$, mean ± SD—% of predicted | 85.5 ± (73–95) |
| $FEV_1$ < LLN, % (n) | 34.1 (73) |
| $FEV_1$/FVC, median (IQR) | 0.84 (0.79–0.87) |
| $FEV_1$/FVC < LLN, % (n) (Obstructive Pattern) | 6 (13) |
| TLC, mean ± SD (n)—% of predicted | 83.1 ± 12.3 |
| TLC < LLN, % (n) (Restrictive Pattern) | 50 (108) |
| VR, median (IQR, n)—% of predicted | 79 (69–93, n = 213) |
| VR/CPT, median (IQR, n) | 0.33 (0.28–0.39, n = 213) |
| DLCO, mean ± SD (n)—% of predicted | 77.4 ± 19.1 |
| DLCO < LLN, % (n) | 48 (103) |

Values are presented as % (n), median [IQR] or mean ± SD. *Abbreviations*: CT, computed tomography; DLCO, diffusion capacity for carbon monoxide; $FEV_1$, forced expiratory volume in 1 s; FVC, forced vital capacity; LLN, lower limit of normal; PFTs, pulmonary function tests; TLC, total lung capacity.

Functional results showed that half of the patients had a restrictive pattern and reduced DLCO, almost 40% had low FVC and $FEV_1$, and 5% had an obstructive pattern.

All patients who had both chest CT and lung function results (n = 214) were stratified into two categories according to the tomographic features: lung without COVID-19-related findings (normal chest CT or with CT abnormalities prior to COVID-19) (N = 39) and lung with COVID-19-related findings (N = 175) (Fig 1). Patients with lung CT with COVID-19-related findings had a higher SAPS3, increased need for IMV and tracheostomy during hospitalization, and a longer ICU hospitalization and IMV period than patients with lung CT without COVID-19-related findings. In addition, patients with lung abnormalities associated with COVID-19 had higher FVC, FEV1, and FEV1/FVC than patients with lung abnormalities without COVID-19-related findings. (S4 Table)

Patients who had lung CT with COVID-19-related findings were subdivided into two further groups according to lesion severity: without fibrotic-like changes and with fibrotic-like changes (Fig 1). Patients with fibrotic-like changes were older, had a longer duration of ICU hospitalization, more frequently needed IMV and tracheostomy, had a higher CT score, FVC, $FEV_1$, and $FEV_1$/FVC, and a lower VR than those without fibrotic-like changes. (Table 3 and S5 Table)

**Table 3. Demographic and clinical characteristics of patients with completed chest CT and lung function results stratified by chest CT lesion severity.**

| | Without Fibrotic-Like Changes (N = 89) | With Fibrotic-Like Changes (N = 86) | p-value |
|---|---|---|---|
| **Demographics** | | | |
| Age, median (IQR, n)—yr | 52.7 (43–62.3, n = 89) | 60.7 (52.5–67.2, n = 86) | 0.016 |
| Male, % (n/N) | 49.4 (44/89) | 51.2 (44/86) | 0.88 |
| BMI, median (IQR, n)—kg/m$^2$ | 32.1 (28.3–35.6, n = 89) | 31 (28.3–35.5, n = 86) | 1 |
| **Characteristics in ICU** | | | |
| ICU lenght of stay, median (IQR, n)—d | 9 (6–13, n = 89) | 19.5 (11–35.5, n = 86) | <0.001 |
| SAPS 3 at admission, mean ± SD (n) | 58.6 ± 14.4 (n = 88) | 60.2 ± 14.2 (n = 83) | 0.717 |
| D Dimer 72h, median (IQR, n)—ng/ml | 1518 (846–4023, n = 85) | 1929 (1073.5–4500.2, n = 80) | 0.851 |
| CRP 72h, median (IQR, n)—mg/l | 163 (73.4–277.8, n = 86) | 156.8 (82.6–240.3, n = 81) | 1 |
| Dialysis, % (n/N) | 16.8 (15/89) | 22.1 (19/86) | 1 |
| Tracheostomy, % (n/N) | 3.4 (3/89) | 18.6 (16/86) | 0.003 |
| VAD, % (n/N) | 27 (24/89) | 45.3 (39/86) | 0.037 |
| IMV during hospitalization, % (n/N) | 64 (57/89) | 81.4 (70/86) | 0.034 |
| Duration of IMV, median (IQR, n)—d | 8 (6–11, n = 51) | 12 (7–19, n = 61) | 0.003 |
| **IMV at first 24 hours** | | | |
| Tidal Volume, median (IQR, n) -ml/kg | 6.2 (6–6.8, n = 50) | 6 (5.9–6.7, n = 58) | 0.676 |
| Minute Volume, median (IQR, n) -l/min | 11.5 (9.3–13, n = 50) | 10.5 (9–12, n = 61) | 0.529 |
| Compliance, median (IQR, n)—mlcmH$_2$O−1 | 30 (23.7–43.9, n = 47) | 29.4 (24.6–38, n = 53) | 1 |
| Respiratory rate, median (IQR, n)—rpm | 28.5 (24.2–35.7, n = 50) | 30 (26–35, n = 61) | 1 |
| PEEP, mean ± SD (n)—cmH$_2$O | 9.7 ± 2.2 (n = 49) | 10.1 ± 2.2 (n = 61) | 0.511 |
| Plateau pressure, median (IQR, n)- cmH$_2$O | 23.5 (18.7–26, n = 44) | 22.5 (20–25, n = 54) | 1 |
| Driving pressure, median (IQR, n)—cmH$_2$O | 13 (10–16, n = 45) | 12 (10–14, n = 54) | 1 |
| PaO$_2$/FIO$_2$ median (IQR, n)—% | 154 (108–217, n = 49) | 142 (113–171, n = 61) | 0.494 |
| Compliance ≥ 40, % (n/N) | 27.7 (13/47) | 22.6 (12/53) | 0.646 |
| Plateau pressure ≥ 28, % (n/N) | 13.6 (6/44) | 11.1 (6/54) | 0.763 |
| Driving pressure ≥ 15, % (n/N) | 35.6 (16/45) | 24.1 (13/54) | 0.806 |
| PaO$_2$/FIO$_2$ ≤ 150, % (n/N) | 49 (24/49) | 55.7 (34/61) | 0.565 |
| Time between hospital admission and the CT follow-up, mean ± SD (n)—d | 531.6 ± 10.3 (n = 89) | 541.9 ± 10.3 (n = 86) | 0.483 |
| CT Score at the follow-up, mean ± SD (n) | 7.2 ± 4.7 (n = 89) | 14.25 ± 4.6 (n = 86) | <0.001 |
| **PFTs at the follow-up** | | | |
| FVC, median (IQR, n)—% of predicted | 80 (73–90, n = 89) | 86 (73–93, n = 86) | 0.126 |
| FVC < LLN, % (n/N) | 50.5 (45/89) | 34 (30/86) | 0.046 |
| FEV$_1$, mean ± SD (n)—% of predicted | 83.6 ± 15,4 (n = 89) | 88.4 ± 16 (n = 86) | 0.040 |
| FEV$_1$ < LLN, % (n/N) | 36 (32/89) | 25.6 (22/86) | 0.144 |
| FEV$_1$/FVC, mean ± SD (n) | 0.81 (0.74–0.85, n = 89) | 0.85 (0.81–0.87, n = 86) | 0.003 |
| FEV$_1$/FVC < LLN, % (n) (Obstructive Pattern) | 3.4 (3/89) | 2.3 (2/86) | 1 |
| TLC, median (IQR, n)—% of predicted | 82 (76–89, n = 89) | 82 (74–91, n = 86) | 1 |
| TLC < LLN, % (n/N) (Restrictive Pattern) | 51.7 (46/89) | 50 (43/86) | 0.88 |
| VR, mean ± SD (n)—% of predicted | 80 (71–93, n = 89) | 73.5 (64.2–87, n = 86) | 0.006 |
| VR/CPT, median (IQR, n) | 33.8 ± 8.2 (n = 89) | 32.8 ± 7.3 (n = 86) | 0.414 |
| DLCO, median (IQR, n)—% of predicted | 81.5 (72.2–93, n = 86) | 76.5 (62.7–85.2, n = 84) | 0.057 |
| DLCO < LLN, % (n/N) | 43.5 (37/85) | 52.4 (44/84) | 0.848 |

Values are presented as median [IQR, n] or % (n/N) or mean ± SD (n). *Abbreviations*: BMI, body mass index; COPD, chronic obstructive pulmonary disease; CRP, c-reactive protein; d, days; DLCO, diffusion capacity for carbon monoxide; FEV$_1$, forced expiratory volume in 1 s; FVC, forced vital capacity; FIO$_2$, inspired fraction of oxygen; ICU, intensive care unit. IMV, invasive mechanical ventilation; PaO$_2$, arterial partial pressure of oxygen; PEEP, positive end-expiratory pressure; PFTs, pulmonary function tests; SAPS3, Simplified Acute Physiology Score 3; SD, standard deviation; TLC, total lung capacity; VAD, vasoactive drugs; yr, years.

The ML model showed that the hospitalization variables selected (sex, ICU length of stay, tracheostomy, duration of IMV, and use of vasoactive drugs) could be predictors of CT lesion severity six to twelve months after ICU admission for COVID-19. The observed performance metrics of the ML prediction model, expressed in terms of mean ± standard deviation and 95% Confidence Interval (CI), were as follows: sensitivity, 0.78 ± 0.02 (95% CI [0.76, 0.79]); specificity, 0.79 ± 0.01 (95% CI [0.78, 0.8]); F1-score, 0.78 ± 0.02 (95% CI [0.76, 0.8]); positive predictive rate, 0.78 ± 0.02 (95% CI [0.76, 0.8]); accuracy, 0.78 ± 0.02 (95% CI [0.76, 0.8]); and AUC, 0.83 ± 0.01 (95% CI [0.82, 0.83]). (S1 Appendix)

## 4. Discussion

To our knowledge, this is the largest prospective cohort study of ICU hospitalized COVID-19 survivors to date that has focused on face-to-face assessment of pulmonary alterations. The use of a protective MV protocol to treat ICU hospitalized COVID-19 patients led to increased patient survival, allowing more patients to be followed up [18]. Our results show that 82% of patients remain with COVID-19-related lung CT sequelae for up to twelve months of follow-up. These long-term imaging alterations were associated with restrictive lung impairment and impaired diffusion capacity in 50% of enrolled patients. However, even in those with fibrotic-like changes, the impairment in PFTs was mild. Additionally, male sex, ICU length of stay, duration of IMV, and need for tracheostomy and vasoactive drugs during hospitalization were predictors of CT lesion severity six to twelve months after ICU admission for COVID-19.

The general evaluation showed that COVID-19 ICU patients with initial pulmonary involvement at follow-up had low respiratory compliance, despite no significant impairment in oxygenation. Worse respiratory mechanics in acute respiratory distress syndrome (ARDS) non-COVID were previously associated with impaired production of pulmonary collagen and independently associated with tomographic and physiological abnormalities after ARDS [25]. This suggests that the reduction in respiratory mechanics, in addition to indicating greater severity of acute lung injury, may increase the risk of pulmonary impairment, perhaps due to the high difficulty associated with adjusting protective ventilatory parameters in these patients. However, this difference was slight, and this hypothesis should be confirmed in future studies.

We found that half of the patients with lung CT with COVID-19 abnormalities had restrictive lung impairment and impaired diffusion capacity. Bellan et al. [26] evaluated a cohort of 200 patients one year after COVID-19 discharge. Their results showed that a high chest CT severity score is one of the most relevant factors associated with low DLCO, as this functional impairment may be secondary to the extent of the pulmonary parenchymal lesions. Additionally, these authors showed that the percentage of patients with impaired DLCO showed no functional improvement from 4 to 12 months of follow-up after COVID-19 hospital discharge, which reinforces the hypothesis of chronic functional impairment.

Follow-up studies have demonstrated that lower DLCO is more frequent than lower TLC in survivors of COVID or ARDS [12, 17, 26–28]. Hui et al. [28] evaluated the PFT outcomes of patients 1 year after hospitalization for ARDS showing a post-discharge frequency of 24% of patients with impaired DLCO and only 5% of patients with low TLC. Notably, they also observed that all PFT predicted values (%) were lower in ICU patients than in ward patients. The incidence of restrictive patterns was higher in our study, which could be a consequence of the critical acute phase of the disease in our population, considering that the other studies evaluated a mixed population of non-ICU and few ICU patients.

A recent meta-analysis of parenchymal lung abnormalities following hospitalization for COVID-19 assessed follow-up studies within twelve months and showed that fibrotic sequelae were estimated in 29% of patients, which is consistent with our findings [29]. Ground-glass

opacities and reticulation were considered non-fibrotic lesions because of the belief that resolution during follow-up was possible, whereas traction bronchiectasis and/or architectural distortion were classified as fibrotic-like changes since they are typically considered more definitive changes [20, 26]. Our findings demonstrated that CT lesion severity did not point to a higher functional limitation, since all CT patterns were associated with mild functional impairment. Fortini et al. [30] evaluated a cohort of patients one year after COVID-19 discharge and, showed that functional improvement was not associated with complete tomographic resolution. In addition, Han et al. [31] did not observe differences in PFTs in a cohort of Chinese patients stratified by the presence or absence of fibrotic interstitial lung abnormalities at the one year COVID-19 follow-up. These results indicate that structural recovery may be slower than functional improvement after COVID-19. Therefore, anatomical sequelae seem to have little functional repercussions, indicating that respiratory evaluation after COVID-19 infection should focus more on functional and clinical evaluation rather than imaging.

Our study also revealed that patients with fibrotic-like changes had a more severe acute phase characterized by longer hospital stay and greater need for IMV and tracheostomy than patients without fibrotic-like changes up to twelve months after COVID-19 hospitalization. Thus, we used an ML to identify the predictors of CT lesion severity at follow-up. Our results showed that male sex, ICU length of stay, duration of IMV, need of tracheostomy, and use of vasoactive drugs are risk factors for CT lesion severity six to twelve months after COVID-19 ICU admission. Previous data reinforce our findings, demonstrating that male sex [32] and length of hospital stay [33] were associated with severe CT lesions one year after COVID-19 hospitalization. Invasive respiratory procedures, such as IMV and tracheostomy, have the inherent potential to induce structural and functional damages to the lung due to inadequate pressure or volume [34]. Additionally, both IMV and vasoactive drugs administered during ICU hospitalization have been associated with increased mortality and complications after discharge [35]. Other characteristics and factors identified during the acute phase of COVID-19 that are associated with a higher risk of development of fibrotic pulmonary lesions in the follow-up of COVID-19 patients in previous studies include a higher CT score of lung involvement, use of high-flow oxygen support, duration of mechanical ventilation, obesity, male sex, smoking, diabetes, and higher levels of C-reactive protein, lactate dehydrogenase, D-dimer, and fibrinogen [8, 20, 32, 36, 37]. Additionally, persistent dyspnea and myalgia and higher serum levels of Krebs von den Lungen 6 (KL-6) at follow-up were associated with a greater risk of occurrence of post-COVID pulmonary fibrosis [31, 37, 38].

Our study had several limitations. First, plethysmography was not performed in all patients who underwent chest CT, reducing (12%) the number of patients evaluated by both methods. Such an examination was not feasible in some cases due to patient limitations (tracheostomy, wheelchair use or intellectual difficulties). Second, we did not exclude patients with COPD. Nevertheless, the number of these patients was small (7%) and, data accuracy was not affected. Another limitation was the recruitment period: we enrolled patients six to twelve months after hospital admission. This recruitment period was selected because it occurred during the first wave of the pandemic when there were restrictions to control the virus, and fear drove people away from hospitals. However, the median follow-up time was approximately 7 months, and most patients enrolled were followed up on time without impacting data accuracy. In addition, the study design allowed mainly the most affected patients to reach the stage of performing chest CT. Thus, patients "without COVID-19-related lesion" included patients with normal chest CT and patients with pre-existing lesions unrelated to COVID-19. This fact contributed to these patients having a lower FVC and FEV1 than patients with COVID-19 lesions, not representing an ideal control group (without lesions) for comparison purposes. In addition, there is variability in the definition of long COVID fibrotic-like changes in the scientific literature.

Although previous data have included reticular opacities as indicative of fibrosis, we understand that, at least in some patients, they may be relatively mild and only associated with ground-glass opacities, which could represent an inflammatory process in resolution, especially organizing pneumonia. Therefore, to increase the specificity of tomography as a method of detecting long COVID definitive fibrosis, our group decided to consider well-established imaging findings indicative of fibrosis (not only in long COVID scenarios, but also in idiopathic interstitial pneumonias), including traction bronchiectasis, architectural distortion, and honeycombing (which was not found in our cohort). Finally, this was a single-center study. However, HCFMUSP is the largest university hospital in our country and, has been designated as a reference hospital to treat COVID-19 patients. Thus, during the COVID-19 pandemic, heterogeneous groups of people of different ethnicities from all districts of the metropolitan region of São Paulo city (approximately 21 million inhabitants) were admitted to this hospital [18].

## 5. Conclusion

Our results show that ICU hospitalization due to COVID-19 led to chronic alterations characterized by imaging and functional abnormalities in the respiratory system that could persist for up to twelve months after hospital admission. The high frequency of lung lesions verified was particularly concerning, mainly because severe CT lesions were more frequent in older patients with more comorbidities, who are prone to infections and acute episodes of exacerbation. It could lead to a collapse in Brazil and the worldwide public health system, and it highlights the importance of a longer follow-up to monitor COVID-19 pulmonary consequences. We believe that monitoring these patients is one way to understand the effects of COVID-19 and to create opportunities to establish public policies that would help to relieve the public health system. Our study paves the way for future investigations focusing on practical options to mitigate the consequences of long COVID-19 and highlights the necessity of longer follow-up.

## Supporting information

**S1 Appendix. Supplemental machine learning model (ML) information's.** Classification of computed tomography (CT) lung lesions. ML prediction model: training and validation. Results. Implementations notes. References.
(DOCX)

**S1 Table. Complementary baseline demographic and clinical characteristics of enrolled patients that underwent the general evaluation.** Values are presented as % (n/N). *Abbreviations*: COPD, chronic obstructive pulmonary disease; FIO$_2$, inspired fraction of oxygen; IMV, invasive mechanical ventilation; PaO$_2$, arterial partial pressure of oxygen; PEEP, positive end-expiratory pressure.
(DOCX)

**S2 Table. Demographic and clinical characteristics of patients with signs of pulmonary involvement stratified by completion of chest computed tomography examination.** Values are presented as median [IQR, n] or % (n/N) or mean ± SD (n). *Abbreviations*: BMI, body mass index; COPD, chronic obstructive pulmonary disease; CRP, c-reactive protein; d, days; FIO$_2$, inspired fraction of oxygen; ICU, intensive care unit. IMV, invasive mechanical ventilation; PaO$_2$, arterial partial pressure of oxygen; PEEP, positive end-expiratory pressure; SAPS3, Simplified Acute Physiology Score 3; SD, standard deviation; VAD, vasoactive drugs; yr, years.
(DOCX)

**S3 Table. Demographic and clinical data of patients with signs of pulmonary involvement stratified by completion of lung function examination.** Values are presented as median [IQR, n] or % (n/N) or mean ± SD (n). *Abbreviations*: BMI, body mass index; COPD, chronic obstructive pulmonary disease; CRP, c-reactive protein; d, days; FIO$_2$, inspired fraction of oxygen; ICU, intensive care unit. IMV, invasive mechanical ventilation; PaO$_2$, arterial partial pressure of oxygen; PEEP, positive end-expiratory pressure; PFTs, pulmonary function tests; SAPS3, Simplified Acute Physiology Score 3; SD, standard deviation; VAD, vasoactive drugs; yr, years.
(DOCX)

**S4 Table. Demographic and clinical characteristics of patients stratified by the presence of COVID-19 CT findings.** Values are presented as median [IQR, n] or % (n/N) or mean ± SD (n). *Abbreviations*: BMI, body mass index; COPD, chronic obstructive pulmonary disease; CRP, c-reactive protein; d, days; DLCO, diffusion capacity for carbon monoxide; FIO$_2$, inspired fraction of oxygen; ICU, intensive care unit. IMV, invasive mechanical ventilation; PaO$_2$, arterial partial pressure of oxygen; PEEP, positive end-expiratory pressure; PFTs, pulmonary function tests; SAPS3, Simplified Acute Physiology Score 3; SD, standard deviation; TLC, total lung capacity; VAD, vasoactive drugs; yr, years.
(DOCX)

**S5 Table. Demographic and clinical characteristics of patients with completed chest CT and lung function results stratified by chest CT lesion severity.** Values are presented as % (n/N). *Abbreviations*: COPD, chronic obstructive pulmonary disease; FIO$_2$, inspired fraction of oxygen; IMV, invasive mechanical ventilation; PaO$_2$, arterial partial pressure of oxygen; PEEP, positive end-expiratory pressure.
(DOCX)

## Acknowledgments

We acknowledge the infrastructure support from the HCFMUSP COVID-19 task force (Antonio José Pereira, Elizabeth de Faria, Lucila Pedroso and Marcelo CA Ramos) during the baseline stage of in-hospital data collection and during the setting-up of the follow-up assessments.
*Members of the HCFMUSP Covid-19 Study Group: Adriana L Araújo[1], Aluisio C Segurado[2], Amanda C Montal[3], Anna Miethke-Morais[3], Anna S Levin[4], Beatriz Perondi[3], Bruno F Guedes[5], Carolina Carmo[3], Carolina S Lázari[6], Cassiano C Antonio[7], Clarice Tanaka[8], Claudia C Leite[9], Cristiano Gomes[10], Edivaldo M Utiyama[11], Emmanuel A Burdmann[12], Eloisa Bonfá[3], Esper G Kallas[2], Ester Sabino[13], Euripedes C Miguel[14], Fabio R Pinna[15], Geraldo F Busatto[14], Giovanni G Cerri[16], Heraldo P Souza[17], Izabel Marcilio[18], Izabel C Rios[3], Jorge Hallak[10], José Eduardo Krieger[19], Juliana C Ferreira[7], Julio F M Marchini[20], Larissa S Oliveira[7], Leila Harima[21], Linamara R Batisttella[22], Luis Yu[5], Luiz Henrique M Castro[5], Marcelo C Rocha[23], Marcello M C Magri[24], Marcio Mancini[25], Maria Amélia de Jesus[3], Maria Cassia J M Corrêa[2], Maria Cristina P B Francisco[3], Maria Elizabeth Rossi[3], Marjorie F Silva[3], Marta Imamura[25], Maura S Oliveira[24], Nelson Gouveia[3], Orestes V Forlenza[14], Paulo A Lotufo[6], Ricardo F Bento[27], Ricardo Nitrini[5], Rodolfo F Damiano[14], Roger Chammas[28], Rossana P Francisco[29], Solange R G Fusco[30], Tarcisio E P Barros-Filho[3], Thais Mauad[31], Thaís Guimarães[3], Thiago Avelino-Silva[3], Vilson C Junior[3] and Wilson J Filho. Affiliations: 1 Diretoria Executiva dos LIMs, Faculdade de Medicina da Universidade de São Paulo HCFMUSP, São Paulo, SP, Brasil. 2 Divisao/Departamento de Molestias Infecciosas e Parasitarias, Hospital das Clinicas HCFMUSP, Faculdade de Medicina, Universidade de Sao Paulo, Sao Paulo, SP, BR. 3 Hospital das Clinicas HCFMUSP, Faculdade de Medicina, Universidade de Sao Paulo, Sao Paulo, SP,

BR. 4 Departamento de Moléstias Infecciosas e Parasitárias do Hospital das Clínicas da Faculdade de Medicina da Universidade de São Paulo, São Paulo, Brazil. 5 Departamento de Neurologia, Faculdade de Medicina da Universidade de São Paulo HCFMUSP, São Paulo, SP, Brasil. 6 Divisão de Laboratório Central do Hospital das Clínicas, da Faculdade de Medicina da Universidade de São Paulo, São Paulo, Brazil. 7 Pulmonary Division, Heart Institute (InCor), Hospital das Clínicas, Faculdade de Medicina, Universidade de São Paulo (HCFMUSP), Sao Paulo, SP, Brazil. 8 Hospital das Clínicas da Faculdade de Medicina, University of São Paulo, Brazil; Department of Physiotherapy, Communication Science and Disorders, Occupational Therapy, University of São Paulo, Brazil. 9 Radiology Institute (InRad), Hospital das Clínicas, Faculdade de Medicina, Universidade de São Paulo (HCFMUSP), Sao Paulo, SP, Brazil. 10 Division of Urology, Hospital das Clinicas, University of Sao Paulo Medical School, Sao Paulo, Brazil. 11 Division of General Surgery and Trauma, Department of Surgery, Hospital das Clínicas da Faculdade de Medicina da Universidade de São Paulo (HC/FMUSP), São Paulo, Brazil. 12 Laboratório de Investigação (LIM) 12, Serviço de Nefrologia, Faculdade de Medicina da Universidade de São Paulo, São Paulo, Brazil. 13 Universidade de São Paulo, Instituto de Medicina Tropical de São Paulo, Laboratório de Investigação Médica (LIM 46), São Paulo, São Paulo, Brazil. 14 Departamento e Instituto de Psiquiatria, Hospital das Clínicas da Faculdade de Medicina da Universidade de São Paulo HCFMUSP, São Paulo, SP, Brasil. 15 Division of Otorhinolaryngology, University of São Paulo, Brazil. 16 Departamento de Radiologia, Faculdade de Medicina, LIM/44, Laboratório de Ressonância Magnética em Neurorradiologia Hospital das Clínicas, Faculdade de Medicina da Universidade de São Paulo, São Paulo, SP, Brasil. 17 Departamento de Clínica Médica, Disciplina de Emergências Clínicas, Hospital das Clínicas da Faculdade de Medicina da Universidade de São Paulo HCFMUSP, São Paulo, SP, Brasil. 18 Epidemiological Surveillance Department, Hospital das Clinicas da Faculdade de Medicina da Universidade de Sao Paulo, Sao Paulo, Brazil. 19 Laboratório de Genética e Cardiologia Molecular, Instituto do Coração (InCor), Hospital das Clínicas, Faculdade de Medicina, Universidade de São Paulo (HCFMUSP), São Paulo, SP, Brazil. 20 Emergency Department, Hospital das Clı́nicas da Faculdade de Medicina da Universidade de São Paulo, São Paulo, Brazil. 21 Clinical Director's Office, Hospital das Clínicas, Faculdade de Medicina, Universidade de São Paulo, São Paulo, Brazil. 22 Departamento de Medicina Legal, Etica Medica e Medicina Social e do Trabalho, Faculdade de Medicina da Universidade de São Paulo HCFMUSP, São Paulo, SP, Brasil. 23 Divisao de Clinica Cirurgica III, Hospital das Clinicas HCFMUSP, Faculdade de Medicina, Universidade de Sao Paulo, Sao Paulo, SP, BR. 24 Division of Infectious Diseases, Faculdade de Medicina, Universidade de São Paulo, São Paulo, Brazil. 25 Instituto de Medicina Física e de Reabilitação, Hospital das Clínicas da Faculdade de Medicina da Universidade de São Paulo, Sao Paulo, Brazil. 25 Unidade de Obesidade e Síndrome Metabólica, Disciplina de Endocrinologia e Metabologia, Hospital das Clínicas da Faculdade de Medicina da Universidade de São Paulo HCFMUSP, São Paulo, SP, Brasil. 26 Centro de Pesquisa Clínica e Epidemiológica, Hospital Universitário, Universidade de São Paulo, São Paulo, Brazil. 27 Divisão de Otorrinolaringologia, Hospital das Clínicas da Faculdade de Medicina da Universidade de São Paulo HCFMUSP, São Paulo, SP, Brasil. 28 Centro de Investigação Translacional em Oncologia, Instituto do Câncer do Estado de São Paulo, Faculdade de Medicina da Universidade de São Paulo, São Paulo, Brazil. 29 Department of Obstetrics and Gynecology, University of Sao Paulo Medical School, Sao Paulo, Brazil. 30 Rheumatology Division, Hospital das Clinicas HCFMUSP, Faculdade de Medicina, Universidade de Sao Paulo, Sao Paulo, Brazil. 31 Departamento de Patologia, LIM/05- Laboratório de Poluição Atmosférica Experimental, Hospital das Clínicas da Faculdade de Medicina da Universidade de São Paulo HCFMUSP, São Paulo, SP, Brasil. 32 Division of Geriatrics, Department of Internal Medicine, Faculty of Medicine,

University of São Paulo (USP), São Paulo, Brazil. Lead author email contact: geraldo.busatto@hc.fm.usp.br.

## Author Contributions

**Conceptualization:** Carlos Roberto Ribeiro Carvalho, Michelle Louvaes Garcia, Paula Gobi Scudeller.

**Data curation:** Carlos Roberto Ribeiro Carvalho, Celina Almeida Lamas, Rodrigo Caruso Chate, João Marcos Salge, Marcio Valente Yamada Sawamura, André L. P. de Albuquerque, Marco Antonio Gutierrez.

**Formal analysis:** Celina Almeida Lamas, Daniel Mario Lima, Marco Antonio Gutierrez.

**Funding acquisition:** Marco Antonio Gutierrez.

**Investigation:** Carlos Roberto Ribeiro Carvalho, Rodrigo Caruso Chate, João Marcos Salge, Marcio Valente Yamada Sawamura, Michelle Louvaes Garcia, Paula Gobi Scudeller.

**Methodology:** Carlos Roberto Ribeiro Carvalho, Marco Antonio Gutierrez.

**Project administration:** Carlos Roberto Ribeiro Carvalho, Michelle Louvaes Garcia, Paula Gobi Scudeller.

**Resources:** Carlos Roberto Ribeiro Carvalho.

**Supervision:** Carlos Roberto Ribeiro Carvalho, Michelle Louvaes Garcia, Paula Gobi Scudeller.

**Validation:** Carlos Roberto Ribeiro Carvalho, Rodrigo Caruso Chate, João Marcos Salge, Marcio Valente Yamada Sawamura, André L. P. de Albuquerque, Carlos Toufen Junior, Cesar Higa Nomura, Bruno Guedes Baldi.

**Visualization:** Carlos Roberto Ribeiro Carvalho, Celina Almeida Lamas, Rodrigo Caruso Chate, João Marcos Salge, Marcio Valente Yamada Sawamura, André L. P. de Albuquerque, Carlos Toufen Junior, Cesar Higa Nomura, Marco Antonio Gutierrez, Bruno Guedes Baldi.

**Writing – original draft:** Celina Almeida Lamas.

**Writing – review & editing:** Carlos Roberto Ribeiro Carvalho, Rodrigo Caruso Chate, João Marcos Salge, Marcio Valente Yamada Sawamura, André L. P. de Albuquerque, Carlos Toufen Junior, Michelle Louvaes Garcia, Paula Gobi Scudeller, Cesar Higa Nomura, Marco Antonio Gutierrez, Bruno Guedes Baldi.

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
