## [Decision Letter · Decision Letter 0]

17 Oct 2022

PONE-D-22-26441Long-term respiratory follow-up of ICU hospitalized COVID-19 patients: prospective cohort studyPLOS ONE

Dear Dr. Ribeiro Carvalho,

Thank you for submitting your manuscript to PLOS ONE. After careful consideration, we feel that it has merit but does not fully meet PLOS ONE’s publication criteria as it currently stands. Therefore, we invite you to submit a revised version of the manuscript that addresses the points raised during the review process.

Both reviewers raised several concerns, especially regarding data interpretation. The authors should fully respond to their comments in the revised manuscript.

We look forward to receiving your revised manuscript.

Kind regards,

Yu Ru Kou, PhD

Academic Editor

PLOS ONE

3. One of the noted authors is a group or consortium HCFMUSP Covid-19 Study Group. In addition to naming the author group, please list the individual authors and affiliations within this group in the acknowledgments section of your manuscript. Please also indicate clearly a lead author for this group along with a contact email address.

Reviewers' comments:

Reviewer's Responses to Questions

**Comments to the Author**

1. Is the manuscript technically sound, and do the data support the conclusions?

Reviewer #1: Yes

Reviewer #2: Partly

2. Has the statistical analysis been performed appropriately and rigorously? 

Reviewer #1: Yes

Reviewer #2: Yes

3. Have the authors made all data underlying the findings in their manuscript fully available?

Reviewer #1: Yes

Reviewer #2: Yes

4. Is the manuscript presented in an intelligible fashion and written in standard English?

Reviewer #1: No

Reviewer #2: Yes

5. Review Comments to the Author

Reviewer #1: Title: Long-term respiratory follow-up of ICU hospitalized COVID-19 patients: prospective cohort study

The authors conducted a prospective cohort study to investigate long-term consequences of COVID-19 on the respiratory system of patients discharged from hospital ICU and to identify risk factors associated with chest CT lesion severity. The authors declared that the ML accurately detected that male sex, ICU and invasive mechanic ventilation (IMV) period, tracheostomy and vasoactive drug need during hospitalization were predictors of CT lesion severity.

Comments:

1. In Abstract, Method, the follow-up period in the study was 6-12 months after hospital discharge. What is the reason for this prospective cohort study designed as an uncertain follow-up period? I suppose that the predictors for fibrotic change of CT checked at 6month may be different from those at 12month due to regression of image abnormalities in COVID-19 patients.

2. In Abstract, Method, the authors did not clearly define the primary outcome measure, CT severity, in this study.

3. In Abstract, Method, the authors did not mention the time point after discharge for performing chest CT and pulmonary function test.

4. In Abstract, Results, that authors mentioned that “An association between the CT feature severity and an impaired PFTs was not found.” However, in Table 3, there were statistical significances in some variables.

5. In Abstract, Conclusion, the authors mentioned “….. were risk factors to the development of severe CT lesion after discharge.”. How to define the development of CT lesion after discharge or during hospitalization? Does it mean that CT lesion got progressive after discharge?

6. In Methods, 2.2 follow-up protocol, the enrolled criteria for performing chest CT and pulmonary function test were different from the description in Abstract. Please reedit it in Abstract.

7. In Results, Table 3, the follow-up duration in each group was not showed here.

8. In Results, Table 3, the authors mentioned that “patients with fibrotic-like changes were older, had a greater duration of ICU hospitalization and need of IMV, were more tracheostomized, had a lower FVC and VR, and a higher FEV1 and FEV1/FVC than those with mild/moderate lesion”. What is the meaning for mild/moderate lesion? The authors did not define what is the severity graded via the CT findings in ICU hospitalized COVID-19 patient in Methods. The authors did not provide CT extent scores, so how to define the severity in fibrotic group (Traction bronchiectasis, but no honeycombing in the study population) greater than non-fibrotic group (GGO,…).

9. In Results, Table 3, what is the reason that the authors did not include the signs of pulmonary involvement (enrolled criteria) as the variables for statistical analysis to test whether the signs of pulmonary involvement were the predictors of CT severity in the study patients?

10. In Discussion, 2nd paragraph, the authors mentioned that “This fact suggests that the respiratory mechanics is possibly more relevant than gas exchange in determining COVID-19 late respiratory alterations.” The authors should have more explanation for this statement. Dose the result of this study support this finding? Moreover, the reference 25 and the conclusion that “these data triggered the idea that possible interventions that improve lung mechanics would be important to reduce these dysfunctions.” Have any evidence to support it and how to do?

11. In Discussion, paragraph 2-4, the authors had discussions about parameters of lung mechanics (e.g., DLCO, TLC) in COVID-19. However, pulmonary function parameters were not the clinical predictors for CT severity in ICU hospitalized COVID 19. in additional to respiratory parameters, the authors can make a more comprehensive literature review to point out the predictors or factor associated with fibrotic-like changes in COVID 19 patients.

Reviewer #2: This manuscript was designed to investigate the long-term respiratory effect, including pulmonary function and chest CT findings, caused by the SARS-CoV-2 infection. Although this research was a prospective study, it seems that patients’ groups and medical record were divided and collected retrospectively. Therefore, the including and excluding criteria were not defined very clear. Several questions need to be answered.

1. Among patients enrolled for Chest CT scan (N=243), how many patients have chronic lung disease, including fibrotic

disease (reticulation), traction bronchiectasis, architectural distortion or malignancy? Were these patients divided into

fibrotic-like or without fibrotic-like change group?

2. Why did authors define reticulation on chest CT scan into the without fibrotic-like change group?

3. Sometimes, we can see “both” fibrotic-like changes and ground-glass opacities in a “single” chest CT scan. Furthermore,

it is very difficult to define this patient into with or without fibrotic-like group. How many patients in this study

encountered this issue and need to be resolved by consensus?

4. In this study, a machine learning model was developed to precited CT lesion severity. Did the authors validate this model

in other cohort to confirm its reproducibility?

6. PLOS authors have the option to publish the peer review history of their article (what does this mean?). If published, this will include your full peer review and any attached files.

Reviewer #1: **Yes: **Hsin-Kuo Ko

Reviewer #2: No

---

## [Author Response · Author response to Decision Letter 0]

15 Dec 2022

Comments to the Author

Editor comments

Editor: 1. Please ensure that your manuscript meets PLOS ONE's style requirements, including those for file naming. The PLOS ONE style templates can be found at https://journals.plos.org/plosone/s/file?id=wjVg/PLOSOne_formatting_sample_main_body.pdf and https://journals.plos.org/plosone/s/file?id=ba62/PLOSOne_formatting_sample_title_authors_affiliations.pdf

Answer: The manuscript meets all PLOS ONE’s style requirements 

Editor: 2. In your Data Availability statement, you have not specified where the minimal data set underlying the results described in your manuscript can be found. PLOS defines a study's minimal data set as the underlying data used to reach the conclusions drawn in the manuscript and any additional data required to replicate the reported study findings in their entirety. All PLOS journals require that the minimal data set be made fully available. For more information about our data policy, please see http://journals.plos.org/plosone/s/data-availability.

Answer: The data availability statement was were specified. (pages 26, lines 558)

Editor: 3. One of the noted authors is a group or consortium HCFMUSP Covid-19 Study Group. In addition to naming the author group, please list the individual authors and affiliations within this group in the acknowledgments section of your manuscript. Please also indicate clearly a lead author for this group along with a contact email address.

Answer: The list of the individual authors and affiliations within the HCFMUSP Covid-19 Study Group, and the email contact of the lead author for this group were included in the acknowledgments section. (Pages 22, 23 and 24. Line 467)

Editor: 4. Please include captions for your Supporting Information files at the end of your manuscript, and update any in-text citations to match accordingly. Please see our Supporting Information guidelines for more information: http://journals.plos.org/plosone/s/supporting-information.

Answer: The captions for your Supporting Information files were included in the end of the manuscript (page 22. Line 424)

Reviewers Comments

Reviewer 1

Reviewer: 1. In Abstract, Method, the follow-up period in the study was 6-12 months after hospital discharge. What is the reason for this prospective cohort study designed as an uncertain follow-up period? I suppose that the predictors for fibrotic change of CT checked at 6 month may be different from those at 12month due to regression of image abnormalities in COVID-19 patients.

Answer: The protocol of this study, described in detail in Busatto et al 2021, considered a 6-9 months follow-up. However, because it occurred during the first pandemic wave when there were restrictions to control the virus and the fear drove people away from hospitals, the recruitment period was extended. The median follow-up time was 219 days (IQR 206 - 291), approximately 7 months, and the minimum and maximum values of this interval were 161 (± 6 months) and 383 days (± 12 months), respectively. Thus, most patients enrolled were followed-up on time, not impacting data accuracy. Nevertheless, we appreciate and understand your concern about it and added this information as a study limitation (Page 20, 381)

Reviewer 1: 2. In Abstract, Method, the authors did not clearly define the primary outcome measure, CT severity, in this study.

Answer: We appreciate the reviewer’s suggestion. The primary outcome measure and CT severity were clearly defined at the Abstract, Method. (Page 2, line 36)

Reviewer 1: 3. In Abstract, Method, the authors did not mention the time point after discharge for performing chest CT and pulmonary function test.

Answer: We appreciate the reviewer’s suggestion. This information was added at the Abstract, Method. (Page 2, line 36)

Reviewer 1: 4. In Abstract, Results, that authors mentioned that “An association between the CT feature severity and an impaired PFTs was not found.” However, in Table 3, there were statistical significances in some variables.

Answer: We appreciate the reviewer’s suggestion and altered this information at the Abstract, Results (Page 2 line 44). Patients without fibrotic-like changes in the CT scan had lower impairment in pulmonary function tests than those with fibrotic-like tomographic changes. 

Reviewer 1: 5. In Abstract, Conclusion, the authors mentioned “….. were risk factors to the development of severe CT lesion after discharge.”. How to define the development of CT lesion after discharge or during hospitalization? Does it mean that CT lesion got progressive after discharge?

Answer: We appreciate the reviewer’s suggestion and altered this information at the Abstract, Conclusion (Page 3, line 53). Our study comprehends a cross-sectional study that evaluated the CT lesion at a specific time-point after hospital admission for COVID-19 and did not have the intention to discuss the CT lesion longitudinal evolution. However, comparing the cases of patients that had a CT at hospital admission and a CT 6-12 months after hospital admission, we verified patients with total lesion regression, others that maintained the lesion pattern and others that developed fibrotic-like lesion thought this time. Unfortunately, few patients had a CT at hospital admission, making it impossible to include this information in the present study. In the future we will have data regarding CT lesion evolution, which will be further investigated in the next follow-up we will perform 2-years after the acute phase.

Reviewer 1: 6. In Methods, 2.2 follow-up protocol, the enrolled criteria for performing chest CT and pulmonary function test were different from the description in Abstract. Please reedit it in Abstract.

Answer: We appreciate the reviewer’s suggestion and altered this information at the Abstract, Methods. (Page 2, line 33)

Reviewer 1: 7. In Results, Table 3, the follow-up duration in each group was not showed here.

Answer: We appreciate the reviewer’s suggestion and added this information at the Results, Table 3. (Page 15)

Reviewer 1: 8. In Results, Table 3, the authors mentioned that “patients with fibrotic-like changes were older, had a greater duration of ICU hospitalization and need of IMV, were more tracheostomized, had a lower FVC and VR, and a higher FEV1 and FEV1/FVC than those with mild/moderate lesion”. What is the meaning for mild/moderate lesion? The authors did not define what is the severity graded via the CT findings in ICU hospitalized COVID-19 patient in Methods. The authors did not provide CT extent scores, so how to define the severity in fibrotic group (Traction bronchiectasis, but no honeycombing in the study population) greater than non-fibrotic group (GGO,…).

Answer: We appreciate the reviewer’s comments and substituted the term “mild/moderate lesion” by “without fibrotic-like changes” (Page 14, line 267). Also, we provided the CT extent score for these groups (without fibrotic-like changes and with fibrotic-like changes). (Methods - Page 8, lines 174 and table 3)

Reviewer 1: 9. In Results, Table 3, what is the reason that the authors did not include the signs of pulmonary involvement (enrolled criteria) as the variables for statistical analysis to test whether the signs of pulmonary involvement were the predictors of CT severity in the study patients?

Answer: Our intention in this study was to identify risk factors in the acute phase of the disease that could be predictors of the CT alterations identified at the follow-up. Thus, the Machine Learning model (ML) was built considering the variables collected at baseline (during hospitalization) with p<0.05 between two categories of CT lesion severity (without fibrotic-like changes and with fibrotic-like changes). Thus, since the analysis performed to evaluate the signs of pulmonary involvement was performed 6-12 months after hospital admission, we considered that it would be not relevant to include them in the ML.

Reviewer 1: 10. In Discussion, 2nd paragraph, the authors mentioned that “This fact suggests that the respiratory mechanics is possibly more relevant than gas exchange in determining COVID-19 late respiratory alterations.” The authors should have more explanation for this statement. Dose the result of this study support this finding? Moreover, the reference 25 and the conclusion that “these data triggered the idea that possible interventions that improve lung mechanics would be important to reduce these dysfunctions.” Have any evidence to support it and how to do? 

Answer: The results showed that low compliance was significantly associated with pulmonary involvement but not PaO2/FIO2. This data suggests that the reduction in respiratory mechanics, in addition to indicating greater severity of acute lung injury, may be associated with an additional risk of pulmonary impairment, perhaps due to the high difficulty associated with adjusting protective ventilatory parameters in these patients. We agree that this difference was slight and we modified the paragraph as suggested by the reviewer in order to emphasize that this is only a hypothesis and should be confirmed in further studies (Page 17, lines 301)

Reviewer 1: 11. In Discussion, paragraph 2-4, the authors had discussions about parameters of lung mechanics (e.g., DLCO, TLC) in COVID-19. However, pulmonary function parameters were not the clinical predictors for CT severity in ICU hospitalized COVID 19. in additional to respiratory parameters, the authors can make a more comprehensive literature review to point out the predictors or factor associated with fibrotic-like changes in COVID 19 patients. 

Answer: Thanks for your comments. We expanded the discussion regarding other factors during the acute and long-term phases that are associated with an increased risk of developing fibrosing lung lesions in the follow-up of patients with COVID-19 (Page 19. Lines 365).

Reviewer 2

This manuscript was designed to investigate the long-term respiratory effect, including pulmonary function and chest CT findings, caused by the SARS-CoV-2 infection. Although this research was a prospective study, it seems that patients’ groups and medical record were divided and collected retrospectively. Therefore, the including and excluding criteria were not defined very clear. Several questions need to be answered.

Reviewer 2: 1. Among patients enrolled for Chest CT scan (N=243), how many patients have chronic lung disease, including fibrotic disease (reticulation), traction bronchiectasis, architectural distortion or malignancy? Were these patients divided into fibrotic-like or without fibrotic-like change group?

Answer: Patients that had at least one of these CT features (reticulation, traction bronchiectasis, architectural distortion or malignancy) previous to COVID-19 were included in the group of patients “Lung without COVID-19-related findings – Lung with CT features previous to COVID-19” and were not considered in the division into fibrotic-like or without fibrotic-like change group. The detailed description of these patients can be seen at the study flowchart (Figure 1). 

Reviewer 2: 2. Why did authors define reticulation on chest CT scan into the without fibrotic-like change group? 

Answer: Reviewing the literature to date, it is possible to notice that there is some variability regarding the tomographic criteria that have been used by different authors to define post-COVID fibrotic-like changes. Although some publications have included reticular opacities as indicative of fibrosis, we understand that, at least in some patients, they may be relatively mild and only associated with ground-glass opacities, which could represent an inflammatory process in resolution, especially organizing pneumonia (OP). Therefore, in order to increase the specificity of tomography as a method of detecting post-COVID definitive fibrosis, our group decided to consider more well-established imaging findings indicative of fibrosis in the literature (not only in the long-term COVID scenario, but also in idiopathic interstitial pneumonias), including traction bronchiectasis and architectural distortion (honeycombing, another classic tomographic finding of fibrosis, was not found in our cohort).

We understand that some patients with reticular opacities in our cohort, even when not associated with traction bronchiectasis or architectural distortion, may also have developed some degree of post-COVID fibrosis, which will be further investigated in the next follow-up we will perform after 2-years of the disease.

Reviewer 2: 3. Sometimes, we can see “both” fibrotic-like changes and ground-glass opacities in a “single” chest CT scan. Furthermore, it is very difficult to define this patient into with or without fibrotic-like group. How many patients in this study encountered this issue and need to be resolved by consensus? 

Answer: The association of traction bronchiectasis and architectural distortion (the imaging characteristics considered to be more definitive of true fibrosis) with the other post-COVID tomographic findings (ie, ground-glass opacities, parenchymal bands, and reticulation) is frequent. In light of that, and as we decided to be more specific rather than sensitive, whenever traction bronchiectasis and/or architectural distortion were observed on CT scans, patients were categorized as belonging to the “fibrotic-like” group, regardless of the presence or absence of associated ground-glass opacities. Probably because our cohort was composed of patients with severe disease in the acute phase, the post-COVID tomographic findings indicative of “fibrotic-like” changes could be identified in the vast majority of those cases by the thoracic radiologists, allowing the categorization of patients into one of the two groups. 

Reviewer 2: 4. In this study, a machine learning model was developed to precited CT lesion severity. Did the authors validate this model in other cohort to confirm its reproducibility?

Answer: We adopted a cross-validation strategy (Page 9 line 194, S1 Appendix). The cross-validation is a statistical method of evaluating and comparing machine learning algorithms by dividing data into two segments: one used to learn or train a model and the other used to validate the model. In typical cross-validation strategy, the training and validation set must cross-over in successive rounds such that each data point has a chance of being validated against. The basic form of cross-validation is k-fold cross-validation. In k-fold cross-validation, the data is first partitioned into k equally (or nearly equally) sized segments or folds. Subsequently k iterations of training and validation are performed such that within each iteration a different fold of the data is held-out for validation while the remaining k − 1 folds are used for learning. In our experiment, we used 3 folds to distribute training and validation sets. (1)

1. Refaeilzadeh P, Tang L, Liu H. Cross-validation. Encyclopedia of Database Systems. Berlin Springer; 2009. p. 532-8.

---

## [Decision Letter · Decision Letter 1]

3 Jan 2023

Long-term respiratory follow-up of ICU hospitalized COVID-19 patients: prospective cohort study

PONE-D-22-26441R1

Dear Dr. Ribeiro Carvalho,

We’re pleased to inform you that your manuscript has been judged scientifically suitable for publication and will be formally accepted for publication once it meets all outstanding technical requirements.

Kind regards,

Yu Ru Kou, PhD

Academic Editor

PLOS ONE

Additional Editor Comments (optional):

Reviewers' comments:

Reviewer's Responses to Questions

**Comments to the Author**

1. If the authors have adequately addressed your comments raised in a previous round of review and you feel that this manuscript is now acceptable for publication, you may indicate that here to bypass the “Comments to the Author” section, enter your conflict of interest statement in the “Confidential to Editor” section, and submit your "Accept" recommendation.

Reviewer #1: All comments have been addressed

Reviewer #2: All comments have been addressed

2. Is the manuscript technically sound, and do the data support the conclusions?

Reviewer #1: Yes

Reviewer #2: Yes

3. Has the statistical analysis been performed appropriately and rigorously? 

Reviewer #1: Yes

Reviewer #2: Yes

4. Have the authors made all data underlying the findings in their manuscript fully available?

Reviewer #1: Yes

Reviewer #2: Yes

5. Is the manuscript presented in an intelligible fashion and written in standard English?

Reviewer #1: Yes

Reviewer #2: Yes

6. Review Comments to the Author

Reviewer #1: The authors have completely responded to my comments. No further issue about dual publication, research ethics or pulbilication ethics was found.

Reviewer #2: The manuscript was significantly improved after major revision and could be accepted in the present form.

7. PLOS authors have the option to publish the peer review history of their article (what does this mean?). If published, this will include your full peer review and any attached files.

Reviewer #1: **Yes: **Hsin-Kuo Ko

Reviewer #2: No

---

## [Editor Report · Acceptance letter]

10 Jan 2023

PONE-D-22-26441R1 

Long-term respiratory follow-up of ICU hospitalized COVID-19 patients: prospective cohort study 

Dear Dr. Ribeiro Carvalho:

I'm pleased to inform you that your manuscript has been deemed suitable for publication in PLOS ONE. Congratulations! Your manuscript is now with our production department. 

Kind regards, 

on behalf of

Dr. Yu Ru Kou 

Academic Editor

PLOS ONE